# Explainable AI for Mathematics: Proofs as Code with Knowledge Graph and Domain Ontology Support

**A. P. Khalov**[*]
Moscow Institute of Physics and Technology (MIPT), Dolgoprudny, Russia
khalov.a@phystech.edu

**O. M. Ataeva**[†]
MIPT, Dolgoprudny, Russia
FRC "Computer Science and Control" of RAS, Moscow, Russia
oataeva@frccsc.ru

**N. P. Tuchkova**[‡]
FRC "Computer Science and Control" of RAS, Moscow, Russia
ntuchkova@frccsc.ru

## Abstract

Neural theorem-proving systems can generate formal proofs, but they often behave as a "black box". It is unclear which pieces of mathematical knowledge led to success or failure. We present SciLibRU, an infrastructure that materializes Lean 4's Mathlib as an ontology-typed knowledge graph (tens of millions of RDF facts) and links mathematical entities to multimodal representations (code, natural-language text, formulae, and related artifacts) under a shared identifier space. Building on this graph, we enable transparent proof support. Using candidate hints that are retrieved via graph navigation and/or semantic search, and each suggestion is explicitly traceable to concrete Mathlib dependency edges. That makes the evidence chain inspectable by humans. Experiments on miniF2F-Test show that graph-based augmentation substantially improves success on harder problems while remaining training-free and composable with any base prover.

## 1 Introduction

Lean 4 and its library Mathlib provide a setting where proofs are executable programs, propositions are types, and correctness reduces to type-checking – making the library's 213,000 verified declarations a natural substrate for AI-assisted reasoning.

Despite this rigor, modern neural theorem provers (Yin et al., 2025; Wang et al., 2025; ByteDance Seed AI4Math, 2025) often remain opaque. High success rates do not explain which knowledge was used and why a proof attempt failed or succeeded. This is problematic because, in formal mathematics, users want not only correctness but also an inspectable process of truth confirmation to see a clear chain showing how the conclusion is justified.

We address this need by proposing transparent proof support through explainable context delivery built on two ideas: (i) an ontology-typed knowledge graph derived from Mathlib's dependency structure, and (ii) a multimodal representation that links the same mathematical entities across formal code, text, notation, and related artifacts under shared identifiers.

[*]ORCID: 0009-0005-4584-8245
[†]ORCID: 0000-0003-0367-5575
[‡]ORCID: 0000-0001-5357-9640

While Mathlib's dependency information is typically available only as a raw import graph without semantic typing or cross-modal links, the ontology layer adds explicit semantics and domain structure. As a result, retrieved proof hints are not just "relevant": each hint can be traced to specific typed dependency relations between declarations, allowing a human to inspect the evidence path and understand why the system proposes particular lemmas.

We focus on a low-budget setting in which only a few attempts are feasible, and we evaluate with (K=8) attempts per task to reliably estimate the probability that at least one attempt succeeds. This value is constrained by available compute: experiments were conducted on a single RTX 3090 GPU, and a full evaluation run ($\sim$35 000 iterations) required approximately 20 days.

We tackle this problem within the SciLibRU project, whose goal is to build an intelligent assistant for reading and checking scientific papers, matching fragments of an article to verifiable knowledge units (definitions, claims, dependencies) and reconstructing justification chains. In this paper, we develop the method for mathematics module of SciLibRU and propose transparent proof support via explainable context delivery. The key idea is to represent Mathlib inside an ontology-typed knowledge graph and connect each entity to a multimodal representation under shared identifiers.

Our contributions (1) conduct a controlled ablation over eight non-chain-of-thought modes on miniF2F-Test (Zheng et al., 2022) with paired statistical testing; (2) show a strongly asymmetric benefit of graph-augmented context on hard problems; (3) demonstrate that deterministic, pattern-based graph entry points can outperform LLM-generated seeds, improving both speed and quality; and (4) provide a training-free, composable method that can be added at inference time on top of a base prover, while keeping the proof-support signal transparent and inspectable through explicit graph traceability.

The paper is organized as follows. Section 2 reviews proofs-as-code, neural theorem proving, and retrieval-augmented approaches. Section 3 describes the knowledge graph construction and multimodal representation. Section 4 details the eight context delivery modes. Sections 5–6 present the experimental setup and quantitative results. Section 7 provides a detailed case study demonstrating the explainability mechanism. We discuss implications in Section 8, and conclude in Section 9. Nine appendices provide the full pairwise $p$-value matrix, detailed mode descriptions, pattern seed categories, prompt templates, additional examples, confidence intervals, and ablation details.

## 2 Background and Related Work

The Curry–Howard correspondence (Howard, 1980) links logic and computation by identifying propositions with types and proofs with terms. Martin-Löf (Martin-Löf, 1984) extends this to dependent types, enabling propositions that quantify over values. Wadler (Wadler, 2015) gives a modern exposition. Lean 4 (de Moura & Ullrich, 2021) implements this "proofs as code" paradigm as a dependently typed language and theorem prover, and its library Mathlib has become the largest unified collection of formally verified mathematics. A practical consequence is that any candidate proof – human or model-produced – can be *verified* by type-checking, yielding an unambiguous notion of correctness that enables rigorous evaluation.

Neural theorem provers broadly follow either a *tree-search* approach, as in Seed-Prover (ByteDance Seed AI4Math, 2025), or a *whole-proof* approach, as in DeepSeek-Prover-V2 (Yin et al., 2025) and Kimina-Prover (Wang et al., 2025). Recent results include Goedel-Prover (Lin et al., 2025), DeepSeek-Prover-V2-7B (Yin et al., 2025) built on DeepSeekMath (Shao et al., 2024), and Hilbert (Varambally et al., 2025). We focus on the whole-proof setting, where the output is a single proof script that can be read and checked end-to-end.

Retrieval-augmented generation (RAG) (Lewis et al., 2020) has also been adapted to theorem proving, including REAL-Prover (Shen et al., 2025), LeanDojo (Hsiang et al., 2025), Herald (Gao et al., 2025), and LeanRAG (Zhang et al., 2025). Related directions include Mathlib semantic search (Gao et al., 2024; Asher, 2025) and autoformalization methods

such as ATLAS (Liu et al., 2025b). Our approach augments an *already specialized* prover at inference time without extra training, retrieves from a *typed dependency graph* rather than a flat store, and evaluates components with *paired statistical tests* in a controlled ablation.

Structured mathematical knowledge representations range from MathML (World Wide Web Consortium, 2010) and OpenMath (Buswell et al., 2004) to ontologies such as OntoMathPRO (Elizarov et al., 2022) and ontology-based digital library models (Serebryakov & Ataeva, 2021). Broader scholarly frameworks include SPAR (Peroni & Shotton, 2018) and the Open Research Knowledge Graph (Brack et al., 2020); formal benchmarks include miniF2F (Zheng et al., 2022) and related suites (Liu et al., 2025c;a); and AI-for-research perspectives appear in (Chen et al., 2025). Building on this line, we construct a knowledge graph from Mathlib dependencies, with nodes as declarations and edges as typed usage relations that capture how lemmas support theorems.

## 3 ARCHITECTURE

***SciLibRU Ontology and Three-Level Model.*** The infrastructure is built on the SciLibRU ontology – a modular meta-level model designed to describe scientific knowledge objects and the processes of their storage, retrieval, and use in computational pipelines. Although in this work only the mathematics domain is considered, the ontology was designed to be domain-general: new subject domains, representations, or computational components can be added without changing the basic semantic invariants. The ontology implements a three-level separation of knowledge structure. The *interpretation level* describes the abstract meaning of a scientific object independently of its form of expression – for instance, the concept "the fundamental theorem of calculus" as understood by a mathematician. The *representation level* fixes the concrete form of expression of this meaning: formal code in Lean 4, natural language text, symbolic notation, or visual rendering. The *resource level* corresponds to the material or computable carrier of representation, including files, database entries, and executable artifacts. Each mathematical statement exists simultaneously at all three levels, linked by a URI (Uniform Resource Identifier) that serves as the canonical identifier across modalities. This separation realizes the principle of invariance which enables scalability to an arbitrary number of domains, and support for alternative interpretations. The three-level model is detailed in a companion paper (Khalov et al., 2026); here we focus on its role as the foundation for our knowledge graph and multimodal retrieval.

***Knowledge Graph Materialization.*** The knowledge graph is constructed by materializing the Lean 4 Mathlib library into the SciLibRU ontological model through a reproducible multi-stage pipeline (Figure 1). The Mathlib source is compiled by Lean 4, producing intermediate representations (`.decl.json` and `.sym.json` files). A dedicated parser (JIXIA) extracts three data types from these files: string metadata (names, docstrings), Lean formal code (type signatures, proof bodies), and structural dependencies between declarations. Additionally, the hierarchical organization of the library is analyzed, reflecting the thematic division of mathematical knowledge into 660 subclasses of the Domain class, each automatically annotated using an LLM.

All extracted entities and connections are then mapped to the SciLibRU ontological model and materialized as an RDF graph. The key methodological assumption is that formal provability in Mathlib serves as a truth criterion: if a statement $\varphi$ is provable in the formal system, then its interpretation in the mathematical domain is considered true:

$$M_{\text{MathLib}} \vdash \varphi \;\Rightarrow\; I_{\text{Math}}(\varphi) = \text{True} \qquad (1)$$

From the structural dependencies, we materialize several types of directed edges. The two most important for retrieval are $\texttt{usesInType}(A, B)$, indicating that declaration $A$ appears in the type signature of $B$ (i.e., $A$ is part of the *statement* of $B$), and $\texttt{usesInValue}(A, B)$, indicating that $A$ is referenced in the proof body of $B$ (i.e., $A$ is used as a *tactic or lemma* in proving $B$). Additional relations include $\texttt{studiedIn}$ (linking statements to mathematical domains, 180K relations) and $\texttt{isSpecializationOf}$ (linking domains hierarchically).

The resulting knowledge graph is substantially larger than the original Mathlib library: starting from approximately 180K declarations, the materialization produces over 66 million RDF triples with 6.3 million unique subjects, stored in GraphDB. This approximately

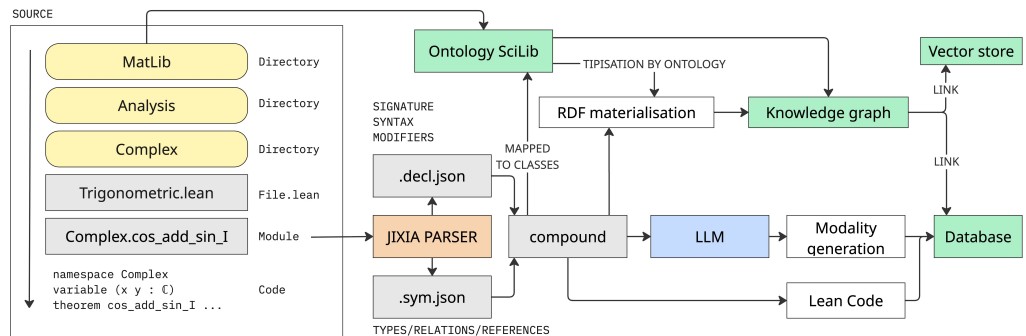

Figure 1: Knowledge graph construction pipeline. Mathlib source is compiled by Lean 4, producing intermediate declaration files. The JIXIA parser extracts declarations, metadata, and structural dependencies, which are mapped to the SciLibRU ontology and materialized as 66M RDF triples in GraphDB.

310-fold increase in expressiveness is achieved through ontological typing, reasoner-based inference that supplements the graph with logically derived triples, and the rich structure of the SciLibRU ontology with its 724 classes described in OWL (Web Ontology Language). The graph is not a direct copy of Mathlib's internal dependency structure as used, for example, in LeanDojo (Hsiang et al., 2025); it is a substantial extension and reinterpretation of formal mathematical statements through a domain ontology that provides semantic context absent from the raw library.

***Multimodal Embedding Layer***. Each Mathlib declaration is represented as a multimodal object: Lean 4 formal code, Russian natural language description, English natural language description, LaTeX symbolic notation, and visual formula rendering (Khalov et al., 2026). Geometrically, a multimodal object is modeled as a simplex in the embedding space, whose vertices correspond to modalities; the centroid serves as an aggregated representation invariant to specific modality (Feng et al., 2014). A dedicated cross-modal encoder, sciLibRuMath, trained on an original dataset of 400,000 mathematical objects, aligns all five modalities into a unified vector space. The resulting embedding layer contains over 1 million embeddings across the five modalities, achieving cross-modal Recall@1 above 90% among 10,000 candidates when searching via centroids for most modality pairs.

## 4 Context Delivery Modes

All modes share the same base model (DeepSeek-Prover-V2-7B) and generation parameters. They differ only in how the prompt is constructed before proof generation. We evaluate 8 modes organized in three groups: one baseline, one vector RAG mode, and six graph RAG variants.

***Baseline and Vector RAG***. The baseline mode **A0** presents the raw Lean 4 goal to the model with no augmentation – only the theorem statement, necessary imports, and a `sorry` placeholder. This measures the model's parametric knowledge alone. Mode **B1** augments the prompt with lemmas retrieved via semantic search: the model generates candidate Mathlib identifiers (up to 20 LLM calls), each of which is embedded and queried against the Qdrant collection via the sciLibRuMath encoder. The top-5 nearest neighbors by cosine similarity are appended to the prompt as unstructured hints before the code fence.

***Graph RAG Variants***. Six graph RAG modes use the Mathlib dependency graph for context retrieval. They differ along three dimensions: *seed selection* (how graph entry points are chosen), *expansion strategy* (how the graph is traversed), and *vector augmentation* (whether semantic search supplements the graph hints).

Two seed selection approaches are compared. *Model seeds* (modes C1, C2) use an LLM call to generate candidate Mathlib identifiers from the goal text, requiring approximately

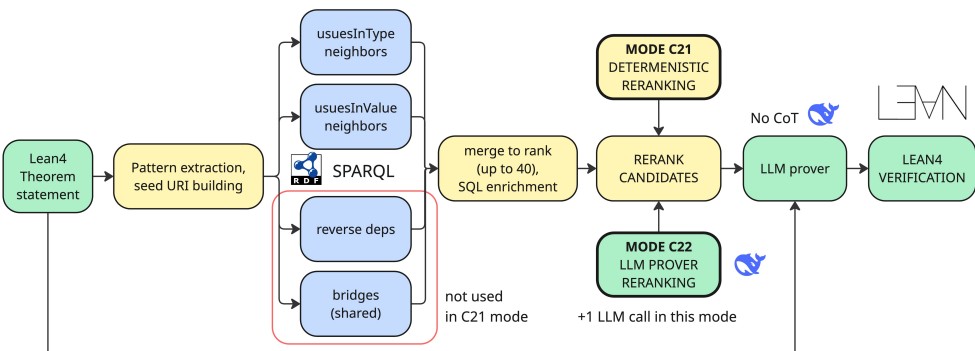

Figure 2: Context delivery pipelines for the two co-leading modes. **C21** (top) combines pattern-based graph expansion with B1 vector retrieval (hybrid). **C23** (bottom) collects candidates from four graph strategies and applies a single LLM reranking call to select the top-8 hints. Both achieve statistically indistinguishable results ($p$=0.95).

30 LLM calls and 130+ seconds per task. *Pattern seeds* (modes C11, C21, C22, C23) use nine regex rules that match goal features – such as inequality operators, `Finset` operations, divisibility predicates, and modular arithmetic symbols – to deterministic Mathlib entry points, requiring zero LLM calls and insignificant overhead.

From the selected seeds, several expansion strategies are evaluated. **C1** expands model-generated seeds via `usesInType` and `usesInValue` edges. **C11** performs the same typed expansion from pattern seeds. **C22** detects bridge nodes – declarations sharing dependencies with the seed set – identifying structurally related lemmas that may not be direct neighbors. **C23** collects candidates from all strategies, then applies a single LLM reranking call (64 tokens, greedy decoding) to select the top-8 hints. Two hybrid modes combine graph and vector retrieval: **C2** merges C1 graph hints with B1 semantic hints (top-3 from each source), and **C21** merges C11 graph hints with B1 semantic hints, leveraging both structural and semantic relevance.

All graph RAG modes format retrieved declarations into four categories reflecting their typical usage in Lean 4 proofs: `apply`/`exact` lemmas applicable as direct proof steps, `rw` lemmas for rewriting equalities, `simp` lemmas for the simplification tactic, and `def` definitions providing vocabulary. This classification makes the retrieved context *actionable*: the model receives not just lemmas, but guidance on *how* to use them. The classification derives from the edge types in the knowledge graph – `usesInType` edges suggest definitional relevance, while `usesInValue` edges suggest proof-body usage. Figure 2 illustrates the pipelines for the two strongest modes.

## 5 EXPERIMENTAL SETUP

We evaluate on miniF2F-Test (Zheng et al., 2022), a cross-system benchmark of 198 formally stated mathematical problems drawn from AMC, AIME, IMO, and MATH competitions, all formalized in Lean 4. As the base model, we use DeepSeek-Prover-V2-7B (Yin et al., 2025) in non-quantized bf16 (bfloat16 floating-point) precision. Generation parameters are held constant across all modes: temperature $T$=0.6, top_p=0.95, top_k=40, max_new_tokens=8192, with each task attempted $K$=8 times per mode.

We define **hard tasks** as those where the unaugmented baseline A0 achieves pass@1 $\leq 25\%$ (at most 2 out of 8 attempts succeed). This yields 109 hard tasks out of 198 total (55%). The threshold is not arbitrary: it identifies tasks where the model's parametric knowledge alone is insufficient. Crucially, it correlates with recognized mathematical difficulty – 91% of AIME problems (10/11), 84% of IMO problems (16/19), and 76% of AMC12 problems (26/34) fall into this category, while only 34% of MATH-D textbook exercises (36/105) do. The threshold thus acts as a principled, model-grounded proxy for mathematical difficulty.

Table 1: Main results on miniF2F-Test. **All tasks** (198): differences are small and not significant. **Hard tasks** (109, A0 pass@1 $\leq$ 25%): all augmentation modes significantly outperform A0.

| Mode | All tasks (198) | | Hard tasks (109) | | | |
|---|---|---|---|---|---|---|
| | pass@1 | pass@8 | pass@1 | $\Delta$ A0 | $p$-value | pass@8 |
| A0 | 41.0 | 53.8 | 3.56 | – | – | 16.5 |
| B1 | 41.0 | 55.8 | 7.34 | +3.78 | 0.0016 | 20.2 |
| C1 | 40.9 | 54.3 | 6.08 | +2.52 | 0.0135 | 18.4 |
| C11 | 42.0 | 52.0 | 8.91 | +5.32 | 0.0050 | 16.7 |
| C2 | 41.0 | 54.6 | 8.10 | +4.51 | 0.0022 | 19.4 |
| **C21** | **43.3** | 54.1 | **10.30** | **+6.71** | **0.0005** | 19.4 |
| C22 | 43.6 | 52.0 | 8.68 | +5.09 | 0.0075 | 15.7 |
| **C23** | **43.4** | **54.6** | **10.42** | **+6.83** | **0.0011** | **21.3** |

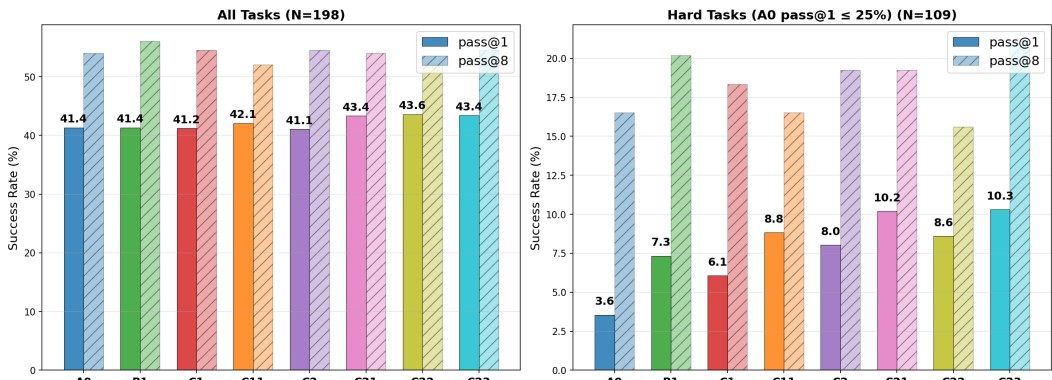

Figure 3: Pass@1 and pass@8 across all modes, stratified by task difficulty. *Left*: on all 198 tasks, modes are indistinguishable. *Right*: on 109 hard tasks, graph RAG modes (C11–C23) substantially outperform the baseline A0. The asymmetry is the key finding.

## 6 RESULTS

Table 1 presents the primary results across all 8 modes, stratified by task difficulty. On all 198 tasks, differences between modes are small (41.0–43.6% pass@1) and not statistically significant. The picture changes dramatically on hard tasks: every augmentation mode significantly outperforms the baseline A0 (all $p < 0.05$), with the best modes nearly tripling the baseline success rate.

***Asymmetric Effect***. The central finding is a strongly asymmetric effect (Figure 3). On all tasks, improvements are not statistically significant ($p > 0.05$). On hard tasks, the same modes yield +2.5 to +6.8 percentage points, all highly significant ($p < 0.05$). This pattern is consistent with the retrieval-augmentation hypothesis: external knowledge is most valuable when the model's parametric knowledge is insufficient. On easy tasks, the model already has adequate knowledge; additional context is either redundant or occasionally distracting. On hard tasks the structured hints from Mathlib's dependency graph provide the missing vocabulary of relevant tactics and lemmas that the model cannot recover from its parameters alone.

***Pattern Seeds vs. Model Seeds***. Regex-based pattern seeds (C11) achieve pass@1 of 8.91% on hard tasks, significantly outperforming LLM-generated model seeds (C1: 6.08%) with $p$=0.05. This result is striking given the cost differential: C11 requires zero LLM calls and runs in 12.1 seconds on average, while C1 requires approximately 30 LLM calls and 133.7 seconds (11 times slower). Their superiority suggests that domain-specific symbolic rules outperform neural text generation for the specific task of selecting entry points into a

structured knowledge graph – a finding with implications for the design of retrieval systems in formal mathematics.

***Co-Leaders: C21 and C23***. The two strongest modes – C21 and C23 – achieve nearly identical hard-task pass@1 (10.30% vs. 10.42%) through fundamentally different mechanisms. C21 combines pattern-based graph expansion with B1 semantic search, providing both structural and semantic context; the added vector retrieval component accounts for its longer running time (38.4s vs. 12.1s for C11 alone). C23 collects candidates from all four graph strategies, then applies a single LLM reranking call (64 tokens) to select the top-8 hints, trading one cheap LLM call for a more curated context.

The Wilcoxon test between C21 and C23 yields $p=0.95$ – they are statistically indistinguishable. The convergence of two independent retrieval strategies to the same performance level strengthens the finding: the effect is robust to the specific graph traversal algorithm and likely reflects the intrinsic value of the underlying Mathlib dependency structure. Notably, C21 has stronger statistical significance against A0 ($p=0.0005$ vs. 0.0011), producing more consistent per-task improvements, while C23 has better pass@8 (21.3% vs. 19.4%) and is 3 faster, producing more diverse correct proofs.

***Coverage Analysis***. Beyond per-task improvement, graph RAG expands the set of solvable tasks. Across all 8 modes combined, 119 out of 198 tasks (60.0%) are solved at least once, compared to 106 (53.8%) for A0 alone. Seven tasks are solved exclusively by graph RAG modes (not by A0 or B1), while only two tasks are solved exclusively by the baseline. This asymmetry in exclusive coverage indicates that graph-augmented retrieval unlocks genuinely new problem-solving capabilities rather than merely increasing the probability of solving already-solvable tasks.

## 7 Case Study: Product Inequality

We illustrate the mechanism of graph-augmented proof generation through a analysis of a hard task: `induction_prod1p1onk3le3m1onn`. This task asks to prove that for all positive integers $n$:

$$\prod_{k=1}^{n}\left(1+\frac{1}{k^3}\right) \leq 3 - \frac{1}{n} \tag{2}$$

The Lean 4 formalization is:

```
theorem induction_prod1p1onk3le3m1onn
  (n : ℕ) (h₀ : 0 < n) : ∏k ∈Finset.Icc 1 n, (1 + (1:ℝ) / k^3) ≤(3:ℝ) - 1 / ↑
      n := by sorry
```

This is a non-trivial inductive inequality involving a finite product over cubic reciprocals. The difficulty lies in the inductive step: splitting the product

$$\prod_{k=1}^{n+1}\left(1+\frac{1}{k^3}\right) = \prod_{k=1}^{n}\left(1+\frac{1}{k^3}\right)\cdot\left(1+\frac{1}{(n+1)^3}\right) \tag{3}$$

and proving that the resulting multiplicative inequality closes correctly with field arithmetic. The contrast between modes is stark: A0 fails all 8 attempts (0/8), while graph-augmented modes succeed up to 75% of the time (C22: 6/8, C21: 5/8, C11: 5/8).

The C21 pipeline extracted pattern seeds from the goal – matching finite product, inequality, and division-by-powers features – and expanded them via `usesInType` and `usesInValue` edges. The structured hints include rewrite lemmas such as $\texttt{div\_le\_iff}_0$ and $\texttt{le\_div\_iff}_0$, which signal multiplicative inequality manipulation patterns; apply lemmas such as `sq_nonneg` and `le_of_eq`, supporting positivity and equality-based steps; and simp set entries such as `pow_nonneg` and `div_self`, directly used by `field_simp` and `positivity` tactics.

The best C21 proof (attempt #57192, verified by Lean 4) uses strong induction, splits the product via `Finset.prod_Icc_succ_top`, and applies `mul_le_mul_of_nonneg_right` to leverage the inductive hypothesis:

```
induction' h₀ with n h₀ ih -- C21 proof (verified):
· norm_num -- Base case n=1:
· cases n with -- Inductive step:
  | zero => contradiction
  | succ n =>
    simp_all [Finset.prod_Icc_succ_top]
    refine' le_trans    -- KEY: use IH on product prefix
      (mul_le_mul_of_nonneg_right
        ih (by positivity)) _
    cases n with
    | zero => norm_num
    | succ n =>
      field_simp
      refine' le_of_sub_nonneg _
      field_simp; ring_nf; positivity
```

The tactic `mul_le_mul_of_nonneg_right` directly corresponds to the retrieved hints `div_le_iff₀` and `le_div_iff₀`, which signal the multiplicative inequality manipulation pattern. The `field_simp; positivity` closure uses `pow_nonneg` and `div_self` from the simp hint set.

Without graph hints, all 8 A0 attempts adopt a fundamentally flawed strategy: bounding each factor $(1 + 1/k^3)$ individually by 2, yielding $\prod \leq 2^n$, then attempting to prove $2^n \leq 3-1/n$ – which is false for $n \geq 2$. The model lacks the vocabulary to recognize that the proof requires strong induction on the product structure and multiplicative inequality manipulation, not per-factor bounding. This example demonstrates the core value proposition: each hint traces to a specific graph edge, and a user inspecting the failed A0 proof alongside the successful C21 proof can identify *exactly which graph-retrieved lemmas* changed the model's strategy. The graph hints systematically corrected the model's reasoning *pattern*: from per-term bounding (a flawed greedy decomposition) to structured induction over the product, a shift that required the vocabulary of multiplicative inequality lemmas unavailable in the model's parametric knowledge.

## 8   DISCUSSION

The two strongest modes (C21, C23) add minimal overhead to the proof generation pipeline. C23 requires a single LLM call of 64 tokens and runs in 12.4 seconds on average – comparable to the unaugmented baseline (11.2s). C21, as a hybrid of graph and vector retrieval, runs in 38.4 seconds (comparable to B1 alone at 38.6s), remaining practical for interactive use. By contrast, model-seed modes (C1, C2) require approximately 30–50 LLM calls and 130–160 seconds, yet achieve *lower* hard-task accuracy, suggesting that inference-time cost and retrieval quality are not positively correlated.

Vector RAG (B1) and graph RAG (C11, C22, C23) solve partially disjoint task sets. The two retrieval paradigms operate on different representations – embeddings for semantic similarity, typed edges for structural dependency – and the 7 graph-exclusive versus 2 baseline-exclusive tasks suggest genuine complementarity. The hybrid modes C2 and C21, which combine both sources, confirm this complementarity: C21 achieves the highest statistical significance against A0 ($p$=0.0005).

The knowledge graph does not "solve" problems – the model does, the graph provides a *vocabulary of relevant tactics and lemmas* that the model lacks when relying on parametric knowledge alone. This framing is key to understanding both the effect size and the explainability claim: the user can inspect the retrieved vocabulary and understand why the model changed its strategy.

While Mathlib covers a substantial portion of modern mathematical knowledge, frontier research problems may require extending the graph with contemporary works that are partially or fully formalized; the SciLibRU architecture supports such extensions, and this effort is currently underway by our research group.

## 9    CONCLUSION

We have shown that structured knowledge retrieval from Mathlib's dependency graph significantly improves neural theorem proving on hard tasks.

The effect is strongly asymmetric: graph augmentation helps substantially on hard tasks ($+6.8$ percentage points, $p<0.001$) while remaining neutral on easy tasks, constraining where retrieval augmentation is useful. Pattern seeds outperform model seeds: deterministic regex-based entry points are 11 times faster and more accurate than LLM-generated ones, demonstrating that simple interpretable rules beat neural generation for knowledge graph navigation. The approach is training-free and composable: no model modification is required, and applying graph-augmented context delivery on top of stronger base models (Lin et al., 2025) is a natural next step.

The key to explainability is traceability: every hint in the augmented prompt maps to a specific edge in the Mathlib dependency graph. This transforms neural theorem proving from a black box into a system where human mathematicians can verify the reasoning behind the automatic proof.

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
