# OpenReview forum: "Explainable AI for Mathematics: Proofs as Code with Knowledge Graph and Domain Ontology Support"
_mathai.club/MathAI/2026/Conference — 2026 Oral_

### Official Review · Reviewer_VTxo · 2026-03-11
**Explainable AI for Mathematics: Proofs as Code with Knowledge Graph and Domain Ontology Support**

**Rating:** 7
**Confidence:** 4

**Review:**

Overall Assessment:
This paper addresses the "black box" problem in current neural theorem proving systems by proposing an innovative, explainable proof-support framework. The core contribution lies in constructing the formal mathematical library (Lean 4 Mathlib) into an ontology-typed knowledge graph linked with multimodal representations, thereby providing traceable context suggestions for the proof generation process. This training-free method significantly improves the proof success rate of the base model on hard problems and enables transparency in the reasoning process. The work is complete, with rigorous experimental design and clear contributions.

Major Strengths:
1.  Precise Problem Identification: Accurately targets a core pain point in AI applications for formal mathematics—the need not only for correctness but also for an explainable proof process. The goal of "transparent proof support" is highly valuable.
2.  Strong Methodological Innovation: The proposed SciLibRU framework is conceptually novel. Its "three-level model" (Interpretation, Representation, Resource) clearly separates different abstraction levels of mathematical objects. Transforming Mathlib's dependency structure into a semantically rich knowledge graph is a key innovation, surpassing flat retrieval stores.
3.  Solid Experimental Design: Conducts systematic controlled ablation studies on a standard benchmark (miniF2F-Test), comparing eight different context delivery modes. Particularly valuable is the definition of a "hard task" subset based on baseline model performance, which reveals a strongly asymmetric effect—graph augmentation provides significant improvement only on tasks where the model's parametric knowledge is insufficient. This offers robust and nuanced evidence for the value of retrieval augmentation.
4.  Notable Practical Findings: Experimental results show that deterministic rule-based pattern seeds outperform LLM-generated model seeds in both effectiveness and efficiency, a finding with direct implications for designing efficient retrieval systems. The proposed methods (C21, C23) dramatically increase the success rate on hard problems with minimal added time overhead, demonstrating practical utility.
5.  Explainability Grounded in Practice: Explainability is not merely a claim but is realized through dependency edges in the knowledge graph. Every hint provided to the model can be traced back to specific entities and relations in the graph, allowing humans to inspect the evidence chain and understand why the system suggests particular lemmas, effectively turning the "black box" into a "white box."

Suggestions for Improvement:
1.  Related Work Section Could Be More Focused: Section 2 (Background and Related Work) is comprehensive but could be slightly streamlined to strengthen the connection to the core method and experiments presented later, focusing more on work most directly related to "explainability" and "structured retrieval."
2.  Case Study Could Highlight Contrast More: The case study in Section 7 effectively demonstrates the mechanism, especially the strategic differences between the baseline and graph-augmented models. The explanation of explainability would be more instructive if it more explicitly summarized how the graph hints systematically correct the model's inherent flawed reasoning patterns (e.g., from "per-term bounding" to "structured induction") rather than just listing the lemmas used.
3.  Discussion of Limitations: The discussion section could more deeply explore the current method's limitations. For instance, how does the method's dependence solely on Mathlib for graph construction affect its coverage and domain bias? How can the approach be extended to mathematical areas not yet formalized in Mathlib?
4.  Figure Readability: As this is a double-blind review version, key figures like Figure 1 (knowledge graph construction pipeline) and Figure 2 (pipelines for the strongest modes) cannot be reviewed within the text. The final version should ensure these figures are clear and self-contained to effectively aid understanding of the complex workflows.

Conclusion:
This is a high-quality, well-executed research paper. It successfully applies knowledge graphs and ontologies to explainable AI for mathematical theorem proving, proposes a novel and effective framework, and rigorously validates its value through experiments. Findings such as the "strongly asymmetric effect" and "pattern seeds outperforming model seeds" hold significant academic and practical value.

---

### Official Review · Reviewer_epgZ · 2026-03-13
**Explainable AI for Mathematics: Proofs as Code with Knowledge Graph and Domain Ontology Support (A Review)**

**Rating:** 7
**Confidence:** 2

**Review:**

The authors present SciLibRU, an infrastructure that materializes Lean 4’s Mathlib as an ontology-typed knowledge graph and links mathematical entities to multimodal representations (code, natural-language text, formulae, and related artifacts) under a shared identifier space. Building on this graph, they enable transparent proof support. Using candidate hints that are retrieved via graph navigation and/or semantic search, and each suggestion is explicitlytraceable to concrete Mathlib dependency edges. It is shown that structured knowledge retrieval from Mathlib’s dependency graph significantly improves neural theorem proving on hard tasks – those where the model’s parametric knowledge is insufficient.

The key to explainability is traceability: every hint in the augmented prompt maps to a specific edge in the Mathlib dependency graph. The user can inspect which lemmas were retrieved, through which graph path, and how they influenced the generated proof.
This transforms neural theorem proving from a black box into a system where human mathematicians can verify not just the proof, but the reasoning behind its construction.

---

### Official Review · Reviewer_ZQP3 · 2026-03-13
**7/10, Major revisions. Impressive infrastructure engineering, but "explainability" framing is misleading (it's transparent retrieval, not XAI), single benchmark, and critical details deferred to unavailable companion paper.**

**Rating:** 7
**Confidence:** 4

**Review:**

This is an impressive infrastructure. The engineering effort is substantial, and the controlled ablation across eight context delivery modes with paired statistical testing demonstrates methodological rigor. A particularly valuable finding is that deterministic pattern-based seeds outperform LLM-generated seeds while requiring zero additional model calls.

The main concern is the framing: the paper claims "Explainable AI," but what is actually demonstrated is transparent retrieval—traceability of retrieved hints to graph edges does not explain the model's internal decision-making process. This distinction matters. Additionally, evaluation is limited to a single benchmark (miniF2F-Test) and a single base model (DeepSeek-Prover-V2-7B). Critical implementation details are deferred to an unpublished companion paper, creating circular reviewing dependencies. Some design choices appear arbitrary (K=8 attempts).

Areas for improvement:
• Reframe "explainability" claims more precisely as "transparent retrieval" or "traceable context delivery"
• Add evaluation on a second benchmark (ProofNet, FIMO, etc.)
• Include essential details from the companion paper or remove the dependency
• Justify the K=8 choice
• Report pattern seed coverage statistics

---

### Decision · Program_Chairs · 2026-03-14

**Decision:**

Accept (Oral)

**Comment:**

Dear Author(s),

On behalf of the Program Committee of the International Conference on Mathematics of Artificial Intelligence (MathAI 2026), we are pleased to inform you that your paper has been accepted for an oral presentation at MathAI 2026.

Your paper was evaluated through a rigorous two-stage review process involving both automated screening and expert review by members of the Program Committee. The reviewers recognized the quality and contribution of your work.

Presentation details:

- Format: Oral presentation (15–20 minutes + 5 minutes Q&A)
- Mode: You may present either in person (offline) at the conference venue in Sirius, Russia, or remotely via Zoom. Please indicate your preferred mode when confirming your participation.
- Conference dates: Marh 30 - April 3, 2026
- Website: https://mathai.club

Next steps:

1. Please confirm your participation and presentation mode by replying to this email mathai.club@yandex.ru no later than March 15, 2026 18:00 Moscow time.
2. If you plan to attend in person, the organizing committee will provide accommodation details separately.
3. Please prepare your final camera-ready manuscript according to the formatting guidelines available at https://mathai.club and upload it to OpenReview by March 15, 2026 18:00 Moscow time.

Should you have any questions regarding the program, logistics, or your presentation slot, please do not hesitate to contact us.

We look forward to your contribution to MathAI 2026.

With kind regards,

MathAI 2026 Program Committee
International Conference on Mathematics of Artificial Intelligence
https://mathai.club
OpenReview: https://openreview.net/group?id=mathai.club/MathAI/2026/Conference
Telegram: https://t.me/MathAI_club
Email: mathai.club@yandex.ru